# Investigation of intrafractional spinal cord and spinal canal movement during stereotactic MR-guided online adaptive radiotherapy for kidney cancer

**Takaya Yamamoto** *, **Shohei Tanaka, Noriyoshi Takahashi, Rei Umezawa, Yu Suzuki, Keita Kishida, So Omata, Kazuya Takeda, Hinako Harada, Kiyokazu Sato, Yoshiyuki Katsuta, Noriyuki Kadoya, Keiichi Jingu**

Department of Radiation Oncology, Tohoku University Graduate School of Medicine, Sendai, Japan

* takaya.yamamoto.c3@tohoku.ac.jp

## Abstract

**Data Availability Statement:** All relevant data are within the manuscript and its Supporting Information files.

### Background and purpose

This study aimed to investigate the intrafractional movement of the spinal cord and spinal canal during MR-guided online adaptive radiotherapy (MRgART) for kidney cancer.

### Materials and methods

All patients who received stereotactic MRgART for kidney cancer between February 2022 and February 2024 were included in this study. Patients received 30–42 Gy in 3-fraction MRgART for kidney cancer using the Elekta Unity, which is equipped with a linear accelerator and a 1.5 Tesla MRI. MRI scans were performed at three points during each fraction: for online planning, position verification, and posttreatment assessment. The spinal cord was contoured from the upper edge of Th12 to the medullary cone, and the spinal canal was contoured from Th12 to L3, using the first MRI. These contours were adjusted to the second and third MR images via deformable image registration, and movements were measured. Margins were determined via the formula "$1.3×Σ+0.5×σ$" and 95% prediction intervals.

### Results

A total of 22 patients (66 fractions) were analyzed. The median interval between the first and third MRI scans were 38 minutes. The mean ± standard deviation of the spinal cord movements after this interval were −0.01 ± 0.06 for the x-axis (right–left), 0.01 ± 0.14 for the y-axis (caudal–cranial), 0.07 ± 0.05 for the z-axis (posterior–anterior), and 0.15 ± 0.08 for the 3D distance, respectively. The correlation coefficients of the 3D distance between the spinal cord and the spinal canal was high (0.92). The calculated planning organ at risk volume margin for all directions was 0.11 cm for spinal cord. The 95% prediction intervals for the x-axis, y-axis, and z-axis were −0.11–0.09 cm, −0.23–0.25 cm and −0.14–0.03 cm, respectively.

**Funding:** TY received the JSPS KAKENHI [Grant Number 24K10879]. URL: https://www.jsps.go.jp/english/ The funder had no role in the study design, data collection and analysis, decision to publish or preparation of the manuscript.

**Competing interests:** TY has received lecturer fees from AstraZeneca KK, Amgen KK, AiRato Inc and Takeda Pharmaceutical Co., Ltd. KJ has received financial support from Elekta KK in the form of a donation. This does not alter our adherence to PLOS ONE policies on sharing data and materials. ST, NT, RU, YS, KK, SO, KT, HH, KS, YK, and NK declare no conflicts of interest relevant to this work.

## Conclusions

Margins are necessary in MRgART to compensate for intrafractional movement and ensure safe treatment delivery.

## Introduction

The concept of planning organ at risk volume (PRV) was introduced in ICRU Report No. 62 to account for its anatomical and geometrical variability [1]. Since then, interfractional anatomical and geometrical movements of the spinal cord and various other sites have been investigated to determine the appropriate PRV margin [2–5]. In the abdominal region, interfractional setup errors are significantly greater than those in the skull, brain and head and neck regions [6]. An investigation in esophageal cancer patients reported that positioning errors in the spinal cord were larger in the abdominal area than in the neck area [2]. Clinically, narrow PRV margins with fiducial marker matching in pancreatic cancer have been associated with higher rates of Grade 3–4 late gastrointestinal toxicity [7].

If interfractional anatomical and geometrical errors were negligible, how should PRV margins be applied to organs? The emergence of MR-Linac has enabled MR-guided online adaptive radiotherapy (MRgART), but the need for PRV margins in MRgART is not well understood. MRI allows for precise visualization of structures such as the spinal cord, which are difficult to identify with electronic portal imaging devices or cone beam CT (CBCT) of conventional Linacs. Consequently, deriving PRV margins from older data is challenging. Nevertheless, we must address a significant concern: intrafractional movement during the longer treatment times required for MRgART. In a human feasibility study of MRgART, the median duration of treatment was 32 minutes [8]. The median durations for the pelvis, upper abdomen, and lung were 46 minutes, ranging from 31 to 113 minutes; 66 minutes, ranging from 38 to 114 minutes; and 41 minutes with a 5-95th percentile ranging from 34 to 58 minutes, respectively [9,10]. Despite the advantage of eliminating interfractional errors, intrafractional movement over such long treatment times must be considered. In this study, we focused on intrafractional movement of the spinal cord and spinal canal, which is not significantly affected by peristalsis, the movement of food, feces and gases, respiration, or medication. Therefore, intrafractional movement of the spinal cord and spinal canal is suitable for determining the need for the PRV margin. We investigated the intrafractional movement of the spinal cord and spinal canal during MRgART of stereotactic radiotherapy (SRT) for kidney cancer.

## Materials and methods

### Patient selection and ethics

Patients who received SRT for kidney cancer via MRgART between February 2022 and February 2024 were identified from the database, which was accessed in March 2024. All of these patients included in the analyses. The authors had access to information that could identify individual participants during and after data collection.

This study was approved by the Ethics Committee of Tohoku University Hospital (reference number: 2023-1-960). Informed consent was waived because of the retrospective design. All patients were guaranteed the opportunity to opt out of participation in this study by receiving information about this study via the internet. Written informed consent as a form of

general consent for the utilization of treatment data was obtained from all patients. All of the methods were performed in accordance with the Declaration of Helsinki.

## SRT procedure, MRI scan and contouring for spinal cord and spinal canal

The MR-Linac used in this study was the Elekta Unity (Elekta AB, Stockholm, Sweden), which is equipped with a 1.5 Tesla MRI scanner and a linear accelerator. Unity 1.5 Tesla MRI and CT scans (SOMATOM Definition AS+, Siemens Medical, Iselin, NJ) were performed for radiotherapy planning. For scanning and radiotherapy, patients were immobilized in the supine position with or without a waist abdominal compression belt (Beruetto; Taketora, Kanagawa, Japan) to control respiratory movement of the tumor. Our MRgART workflow with Elekta Unity has been described previously [11]. In the MRgART workflow, a fat-suppressed T2-weighted MRI scan with a breathing navigator was performed with the following parameters: TR 2100 ms, TE 252 ms, acquired voxel size $2.0 \times 2.0 \times 2.4$ mm$^3$, reconstructed voxel size $0.79 \times 0.79 \times 1.2$ mm$^3$, and FOV 360 (anterior-posterior) × 455 (left-right) × 280 (cranial-caudal) mm$^3$. MRI scans were performed three times: an initial scan for online delineation and planning (the first MRI), a second scan for position verification which was performed immediately before beam-on (the second MRI), and a third scan for posttreatment assessment, which was performed immediately after beam-off (the third MRI). Irradiation for kidney cancer patients was administered at doses of 30–42 Gy in 3 fractions via the intensity-modulated radiotherapy (IMRT) technique (step-and-shoot method) with 7 MV flattening-filter-free photons.

The spinal cord and spinal canal were contoured on the first MRI using Monaco v5.51.11 (Elekta AB, Stockholm, Sweden). In this study, the spinal cord was contoured from the upper edge of Th12 to the medullary cone using the first MRI. The spinal canal was contoured from the upper edge of Th12 to the lower edge of L3, and structures for the entire spinal canal (Th12–L3) as well as each vertebral body level (Th12, L1, L2, L3) were created using the first MRI. If the scanning area was not large enough to delineate the entire spinal canal or spinal cord, delineable parts of the spinal canal were created at each level. All the "transformation data" values were subsequently reset to zero, and the delineations were adjusted to the second and third MR images via deformable image registration (DIR) through the "Adapt Anatomy" function in Monaco. To confirm the accuracy of the DIR, manually rigid registration with or without manual modification of structure contouring (manual registration) was also performed by a radiation oncologist with 14 years of experience.

## Calculation of margins and statistical analysis

Three-dimensional coordinates of each structure were obtained from the "center of structure" option of Monaco; the coordinates consisted of the x-axis, y-axis and z-axis, which were the lateral axis, craniocaudal axis and vertical axis, respectively, with three-dimensional coordinates in centimeters.

The intrafractional movement of each structure was calculated via subtraction between the first and second MR images and between the first and third MR images. The movements of the x-axis, y-axis and z-axis were expressed as X, Y and Z, respectively, and the 3D distance was calculated using the following formula: $\sqrt{X^2 + Y^2 + Z^2}$. To evaluate the correlation, Pearson's correlation coefficients were computed between the 3D distances calculated via DIR and those obtained via manual registration. Additionally, Pearson's correlation coefficients were calculated to assess the discrepancy in 3D distances between the spinal cord and the spinal canal, in order to determine whether the movement of the spinal cord originates from the patient's movement or from spinal cord itself within the spinal canal.

The concept, methodology and formula reported by McKenzie et al. and Suzuki et al. were used for the PRV margin calculation [12,13]. In MRgART with the adapt-to-shape method, which allows radiation oncologists to modify the contouring of the tumor and organs via on-line MRI, interfractional structure movement is modified at the time of the first MRI, effectively reducing interfractional setup margins to zero [14]. Therefore, we calculated the margins solely for the intrafractional movements by analyzing the three-dimensional coordinate shift of each structure point via the "center of structure" option. The intrafractional random error and systematic error were calculated from the intrafractional data, which included the patient-specific factors such as back pain, spinal stenosis, and muscle tension. While continuous and linearly increasing motion allows for margin approximation, the movements of the spinal cord and spinal canal do not drift and are not assumed to follow a linear pattern [15]. Therefore, the first, second, and third MRI were used to calculate the intrafractional random error and systematic error.

1. Intrafractional systematic error ($\Sigma$-intra): The average structural movements at each fraction were calculated by using the 3D distance from the first MRI to the second MRI (Distance1) and from the first MRI to the third MRI (Distance2). The overall mean ± standard deviation (SD) of these average movements for each fraction was then calculated across all patients. The resulting SD is termed SD1, representing $\Sigma$-intra (intrafractional systematic error).

2. Intrafractional random error ($\sigma$-intra): The mean ± SD between Distance1 and Distance2 was calculated for each fraction, and the SD for each fraction was determined (SD2). The root mean square of SD2 across all patients and fractions was then calculated and used as $\sigma$-intra, representing intrafractional random error.

Finally, the PRV margin was calculated using the following formula: $1.3 \times \Sigma$-intra$+0.5 \times \sigma$-intra (hereafter referred to as the PRV margin). For reference, the margin formula for planning target volume (PTV) was also calculated using the following formula: $2.5 \times \Sigma$-intra$+0.7 \times \sigma$-intra [16] (hereafter referred to as the Reference margin).

In addition to these margin formulas, 95% prediction intervals were also calculated via two-sided tests because asymmetrical structural movement can occur [17]. JMP v. 17.1.0 (SAS Institute, Cary, NA) was used for prediction interval calculations.

When the difference between Distance1 and Distance2 for each patient was compared, a two-sided paired t test was performed with EZR v1.54 [18]. The significance level was set to 5%.

## Results

A total of 22 patients were identified, and all patients completed 3-fraction SRT for kidney cancer (Table 1). None of the patients had vertebral lesions, including spine metastases, symptomatic hernias or symptomatic spinal stenosis. The median interval between the first MRI scan and the second MRI scan was 14 minutes (interquartile range 12–18 minutes, range 8–28 minutes), and the median interval between the first MRI scan and the third MRI scan was 38 minutes (interquartile range 34–43 minutes, range 21–61 minutes).

The movement of the centers of the structures via the DIR method is summarized in Table 2 for Distance1 and in Table 3 for Distance2. Among the three directions, the absolute mean value of the z-axis (vertical direction) was the highest, indicating a preference for movement in the posterior direction. In contrast, the SD of the y-axis (craniocaudal direction) was the highest, indicating relatively large movement in the craniocaudal direction without a preference for the cranial or caudal direction. The 3D distances are also summarized in the tables. The correlation coefficients between DIR and manual registration via Distance1 and Distance2

**Table 1. Patients characteristics and treatment information.**

| Category | Median (range) or distribution |
|---|---|
| Number of patients and treatment | 22 patients and 66 treatments |
| Age | 75 years (47–83 years) |
| Sex | Male: 17, Female: 5 |
| Actual radiotherapy dose | 42 Gy: 20, 30 Gy: 2 |
| Waist belt | Yes: 18, No:4 |
| Vertebral lesion | No: 22 |
| Location of medullary cone | L1: 16, L2: 5, L3: 1* |
| Intervals between the first MRI and the second MRI | 14 minutes (8–28 minutes) |
| Intervals between the first MRI and the third MRI | 38 minutes (21–61 minutes) |
| Intervals between beam on and beam off | 12 minutes (8–23 minutes) |

*Patient whose medullary cone was located at the L3 level had six lumbar vertebrae.

were 0.76 (p<0.01) and 0.83 (p<0.01) for the spinal cord and 0.89 (p<0.01) and 0.88 (p<0.01) for the spinal canal, respectively. The correlation coefficients of the 3D distance from the DIR between the spinal cord and the spinal canal using Distance1 and Distance2 were 0.88

**Table 2. Movement of the center of the structures between the first and second MR images with deformable image registration.**

| | Axis* | No. | Mean (cm) | SD | Range (cm) |
|---|---|---|---|---|---|
| *Spinal canal* | X | 57† | <±0.01 | 0.04 | −0.10, 0.09 |
| *(Th12–L3)* | Y | | <±0.01 | 0.09 | −0.36, 0.12 |
| | Z | | −0.02 | 0.02 | −0.09, 0.04 |
| | 3D distance | | 0.08 | 0.06 | 0.01, 0.36 |
| *Spinal cord* | X | 59 | <±0.01 | 0.04 | −0.12, 0.07 |
| *(Th12–cone)* | Y | | 0.01 | 0.10 | −0.36, 0.12 |
| | Z | | −0.04 | 0.03 | −0.12, 0.02 |
| | 3D distance | | 0.10 | 0.07 | 0.01, 0.36 |
| *Th12* | X | 61 | <±0.01 | 0.04 | −0.10, 0.07 |
| | Y | | 0.01 | 0.10 | −0.36, 0.12 |
| | Z | | −0.03 | 0.03 | −0.13, 0.02 |
| | 3D distance | | 0.09 | 0.07 | 0.01, 0.36 |
| *L1* | X | 61 | <±0.01 | 0.04 | −0.10, 0.08 |
| | Y | | <±0.01 | 0.10 | −0.36, 0.12 |
| | Z | | −0.02 | 0.02 | −0.11, 0.05 |
| | 3D distance | | 0.09 | 0.07 | 0.01, 0.36 |
| *L2* | X | 63 | <±0.01 | 0.04 | −0.12, 0.08 |
| | Y | | <±0.01 | 0.09 | −0.36, 0.12 |
| | Z | | −0.02 | 0.03 | −0.09, 0.05 |
| | 3D distance | | 0.08 | 0.07 | 0.01, 0.36 |
| *L3* | X | 61 | <±0.01 | 0.04 | −0.13, 0.11 |
| | Y | | −0.01 | 0.09 | −0.36, 0.12 |
| | Z | | −0.02 | 0.03 | −0.10, 0.06 |
| | 3D distance | | 0.08 | 0.07 | 0.01, 0.36 |

*Positive directions of the x-axis (lateral axis), y-axis (craniocaudal axis) and z-axis (vertical axis) are the left, cranial and anterior directions, respectively. †The number 57 indicates that 57 MRI series included the scanning area from the upper edge of Th12 to the lower edge of L3 in both the first and second MRIs, and the spinal canal movement in these 57 MRI series was measured.

Abbreviations: No.: Number of measurable MRI series; SD: Standard deviation.

**Table 3. Movement of the center of the structures between the first and third MR images with deformable image registration.**

|  | Axis | No. | Mean (cm) | SD | Range (cm) |
|---|---|---|---|---|---|
| *Spinal canal* | X | 52 | <±0.01 | 0.06 | −0.14, 0.10 |
| *(Th12–L3)* | Y |  | −0.01 | 0.13 | −0.42, 0.18 |
|  | Z |  | −0.05 | 0.05 | −0.14, 0.09 |
|  | 3D distance |  | 0.13 | 0.09 | 0.02, 0.43 |
| *Spinal cord* | X | 55 | −0.01 | 0.06 | −0.14, 0.09 |
| *(Th12–cone)* | Y |  | <±0.01 | 0.14 | −0.42, 0.24 |
|  | Z |  | −0.07 | 0.05 | −0.19, 0.05 |
|  | 3D distance |  | 0.15 | 0.08 | 0.02, 0.42 |
| *Th12* | X | 56 | <±0.01 | 0.06 | −0.13, 0.10 |
|  | Y |  | 0.01 | 0.13 | −0.42, 0.24 |
|  | Z |  | −0.05 | 0.04 | −0.17, 0.06 |
|  | 3D distance |  | 0.14 | 0.08 | 0.01, 0.42 |
| *L1* | X | 58 | <±0.01 | 0.05 | −0.14, 0.10 |
|  | Y |  | <±0.01 | 0.14 | −0.48, 0.18 |
|  | Z |  | −0.05 | 0.05 | −0.15, 0.10 |
|  | 3D distance |  | 0.14 | 0.10 | 0.02, 0.49 |
| *L2* | X | 60 | <±0.01 | 0.05 | −0.16, 0.10 |
|  | Y |  | −0.01 | 0.14 | −0.48, 0.18 |
|  | Z |  | −0.05 | 0.06 | −0.19, 0.16 |
|  | 3D distance |  | 0.13 | 0.10 | 0.03, 0.50 |
| *L3* | X | 57 | <±0.01 | 0.06 | −0.21, 0.13 |
|  | Y |  | −0.02 | 0.13 | −0.48, 0.18 |
|  | Z |  | −0.04 | 0.06 | −0.20, 0.18 |
|  | 3D distance |  | 0.14 | 0.10 | 0.02, 0.51 |

Abbreviations are the same as those in Table 2.

(p<0.01) and 0.92 (p<0.01), respectively. As a result, it was determined that the movement of the spinal cord originates from the patient's movement. Additionally, the individual values of 3D distances for Distance1 and Distance2 of the spinal cord for each patient are plotted in Fig 1A and 1B, respectively. The overall mean of the average 3D distance across 3 fractions for each patient was 0.10 cm for Distance1 and 0.15 cm for Distance2 (p<0.01). The mean variance of the 3-fraction 3D distance for each of the 22 patients was 0.0021 cm for Distance1 and 0.0010 cm for Distance2 (p = 0.33). As shown in Fig 1, two patients (No. 10 and No. 18) exhibited larger values derived from spinal cord. The mean movements of Distance2 for Patient No. 10 along the x-axis, y-axis, z-axis, and 3D distance were 0.01 cm, −0.26 cm, −0.09 cm and 0.28 cm, respectively. For Patient No. 18, the corresponding values were 0.01 cm, −0.42 cm, 0.01 cm and 0.42 cm, respectively. Patient No. 10 reported increasing back pain over time while in the supine position, while no specific reason could be identified for the other patient. Examples of the MR images of these patients are shown in Figs 2 and 3.

The results of the PRV and Reference margin calculations are shown in Table 4. The calculated PRV margin for all directions of expansion was 0.11 cm for the spinal canal and the spinal cord. When asymmetrical PRV margins were considered, the 95% prediction intervals for the x-axis (positive direction is left), y-axis (positive direction is cranial), and z-axis (positive direction is anterior) were −0.10–0.10 cm, −0.23–0.22 cm and −0.12–0.04 cm for the spinal canal, respectively, and −0.11–0.09 cm, −0.23–0.25 cm and −0.14–0.03 cm for the spinal cord, respectively.

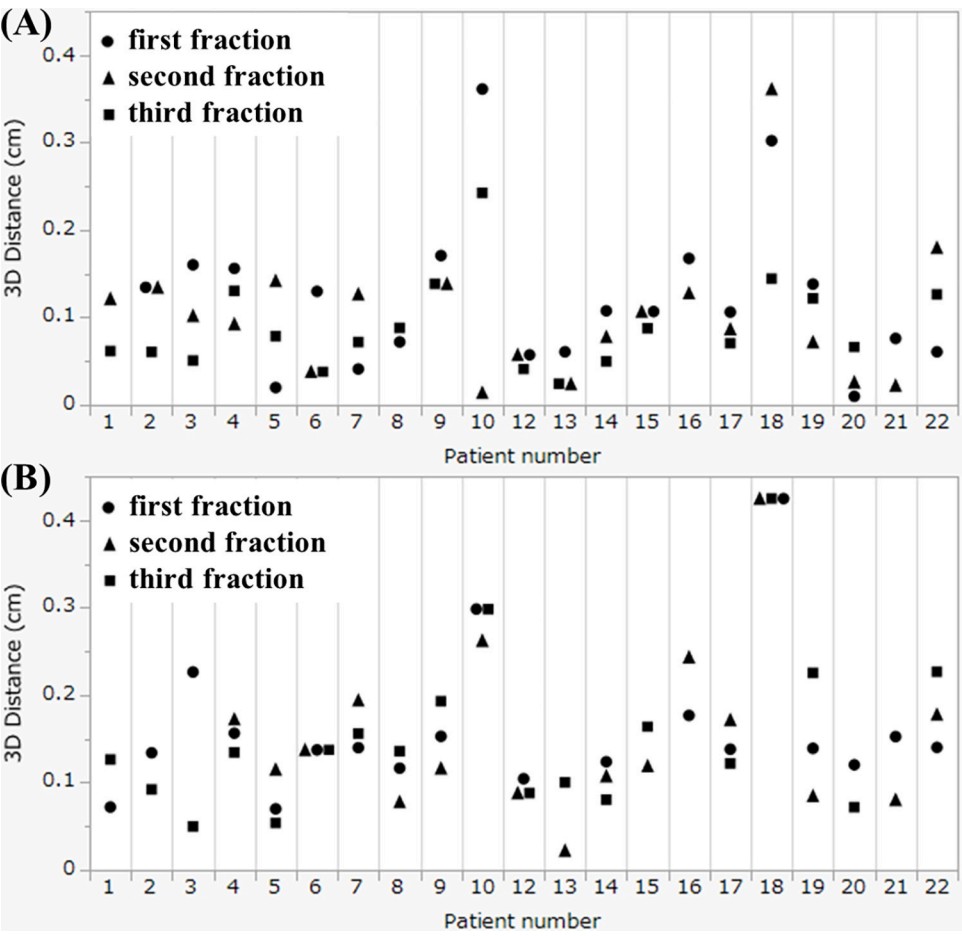

**Fig 1. Individual 3D distance data of intrafractional movement values derived from spinal cord.** (A) shows the movement from the planning MRI scan to the position verification MRI scan, and (B) shows the movement from the planning MRI scan to the posttreatment assessment MRI scan. The movements during the first, second, and third treatment fractions are plotted using circles, triangles, and squares, respectively.

## Discussion

This study revealed intrafractional movement and recommended margins during MRgART. The strength of this study was the use of MRI for delineation. Unlike previous studies that relied on bony landmarks, MRI enabled the direct contouring of the spinal cord and spinal canal. From this perspective, this study is the first to directly measure the movement of these structures during radiotherapy [13,19]. As a result, the movement of the spinal cord itself within the spinal canal was not observed, and the movement of the spinal cord resulted from the patient's movement. Our motivation was to reduce the PRV margin to 0 cm using MRgART with the adapt-to-shape method because the spinal cord and spinal canal are minimally affected by respiratory motion and organ movement during MRgART. However, the results indicated that there was some movement originated from the patient's movement, including muscle relaxation or postural adjustments. According to dosimetric analysis of reirradiation via stereotactic radiotherapy (SRT) for the thoracic spine, MRgART with the adapt-to-shape method using on-line MRI increases the minimum dose to the gross tumor volume (GTV) in most cases while maintaining spinal cord doses similar to those in reference radiotherapy plans (clinical plans) [20]. In contrast to the adapt-to-shape method, previous studies

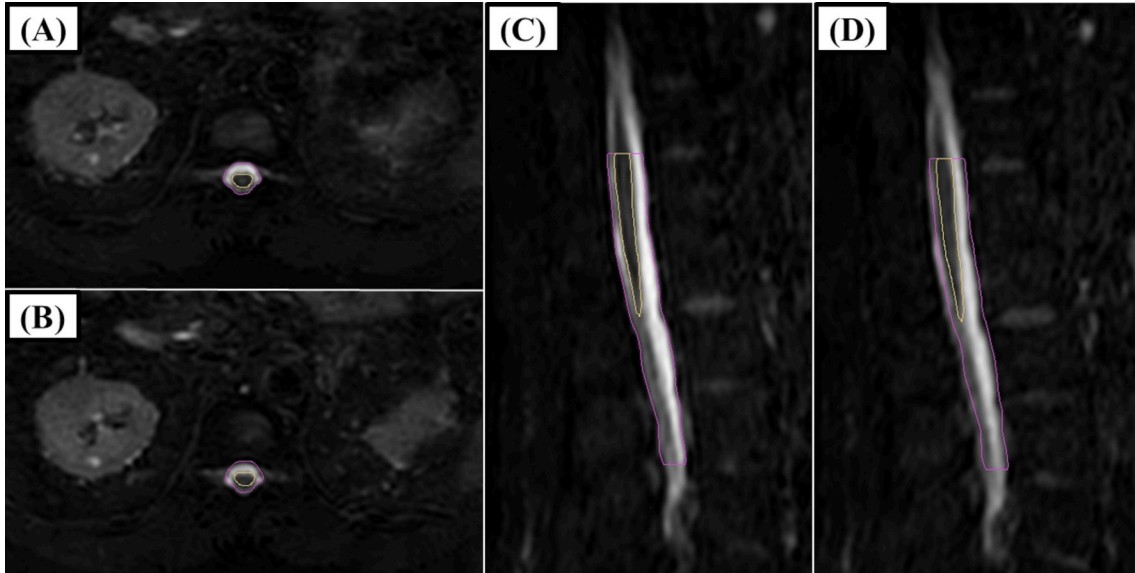

**Fig 2. An example of an MRI image of patient no. 10.** A and C show the contouring of the spinal cord (yellow) and spinal canal (pink) via the first MRI. B and D show the third MR image with the contouring of the spinal cord and spinal canal on the first MR image. On the third MRI (B and D), movement in the caudal and posterior directions was observed. There was relatively large movement in the caudal direction; however, a relatively small portion was out of the contour due to the cylindrical shape of the structures.

reported that all plans using the adapt-to-position method, which provides only a shift of the plan to the patient using on-line MRI, presented a 10% or greater increase in the near maximum dose to the spinal cord. Overall, even with MRgART, it was difficult to reduce the PRV margin to 0 cm.

Because bony anatomical matching has long been in use, investigations of the PRV margin and clinical evidence for the PRV margin for the spinal cord have already been reported. Suzuki et al. reported PRV margins of 0.22–0.24 cm for the cervical vertebrae, which were calculated from interfractional and intrafractional errors [13]. They measured intrafractional

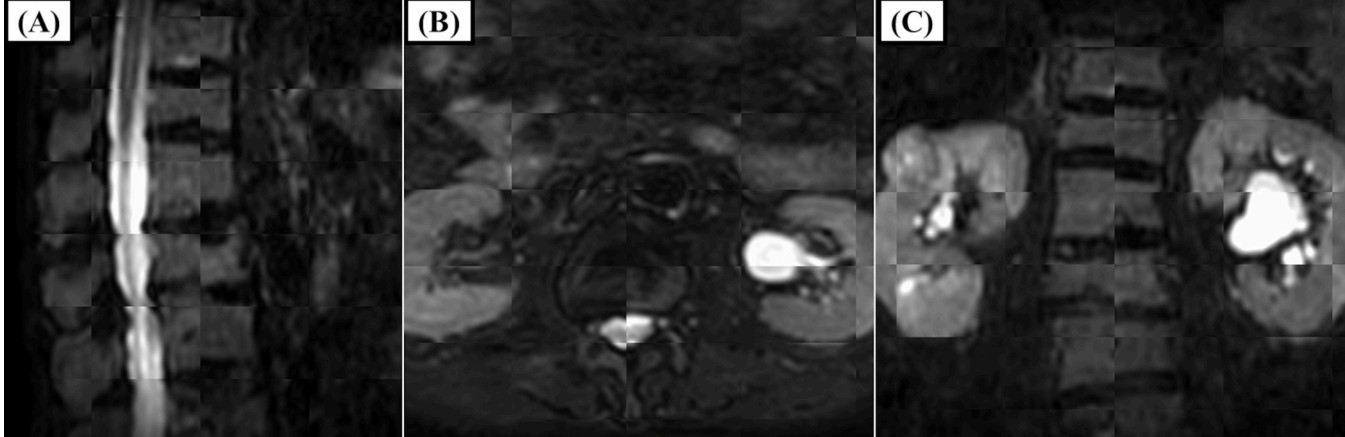

**Fig 3. MRI images of patient no. 18.** The sagittal, axial, and coronal views in a checkerboard format comparing the first and third MRI scans are shown in A, B, and C, respectively. C shows that the movement of the vertebrae in the craniocaudal direction was easy to observe, and relatively large movements were observed in this direction.

**Table 4. The result of planning organ at risk volume margin calculation from the movement of the center of structures via MRI for adaptive planning, position verification and posttreatment assessment with deformable image registration.**

| | Axis | No. | Mean (cm) | SD | Range (cm) | 95% prediction intervals* | PRV / Reference margin from the formula |
|---|---|---|---|---|---|---|---|
| Spinal canal | X | 109 | <±0.01 | 0.05 | −0.14, 010 | −0.10, 0.10 | 0.07 / 0.13 |
| (Th12–L3) | Y | | <±0.01 | 0.11 | −0.42, 0.18 | −0.23, 0.22 | 0.16 / 0.30 |
| | Z | | −0.03 | 0.04 | −0.14, 0.09 | −0.12, 0.04 | 0.11 / 0.20 |
| | 3D | | 0.11 | 0.08 | 0.01, 0.43 | 0.25 | 0.11 / 0.21 |
| Spinal cord | X | 114 | <±0.01 | 0.05 | −0.14, 0.09 | −0.11, 0.09 | 0.07 / 0.14 |
| (Th12–cone) | Y | | <±0.01 | 0.12 | −0.42, 0.24 | −0.23, 0.25 | 0.17 / 0.32 |
| | Z | | −0.05 | 0.04 | −0.19, 0.05 | −0.14, 0.03 | 0.06 / 0.11 |
| | 3D | | 0.12 | 0.08 | 0.01, 0.42 | 0.26 | 0.11 / 0.22 |
| Th12 | X | 117 | <±−0.01 | 0.05 | −0.13, 0.10 | −0.11, 0.10 | 0.07 / 0.14 |
| | Y | | 0.01 | 0.12 | −0.42, 0.24 | −0.22, 0.25 | 0.17 / 0.31 |
| | Z | | −0.04 | 0.04 | −0.17, 0.06 | −0.12, 0.03 | 0.05 / 0.10 |
| | 3D | | 0.11 | 0.08 | 0.01, 0.42 | 0.25 | 0.11 / 0.21 |
| L1 | X | 119 | <±0.01 | 0.05 | −0.14, 0.10 | −0.10, 0.09 | 0.07 / 0.13 |
| | Y | | <±0.01 | 0.12 | −0.48, 0.18 | −0.25, 0.25 | 0.17 / 0.33 |
| | Z | | −0.04 | 0.04 | −0.15, 0.10 | −0.12, 0.04 | 0.05 / 0.10 |
| | 3D | | 0.11 | 0.09 | 0.01, 0.49 | 0.27 | 0.13 / 0.24 |
| L2 | X | 123 | <±0.01 | 0.05 | −0.16, 0.10 | −0.10, 0.10 | 0.05 / 0.10 |
| | Y | | −0.01 | 0.12 | −0.48, 0.18 | −0.25, 0.22 | 0.17 / 0.32 |
| | Z | | −0.03 | 0.05 | −0.19, 0.16 | −0.14, 0.06 | 0.06 / 0.12 |
| | 3D | | 0.11 | 0.09 | 0.01, 0.50 | 0.26 | 0.13 / 0.24 |
| L3 | X | 118 | <±0.01 | 0.05 | −0.21, 0.13 | −0.10, 0.11 | 0.07 / 0.14 |
| | Y | | −0.01 | 0.11 | −0.48, 0.18 | −0.25, 0.21 | 0.16 / 0.30 |
| | Z | | −0.03 | 0.05 | −0.20, 0.18 | −0.14, 0.07 | 0.07 / 0.13 |
| | 3D | | 0.11 | 0.09 | 0.01, 0.51 | 0.26 | 0.13 / 0.24 |

*Prediction intervals for the x-axis, y-axis and z-axis were calculated via two-sided tests, and those for the 3D distance were calculated via a one-sided test for the upper limit.

Abbreviations: No.: Number of measurable MRI series, SD: Standard deviation, PRV: Planning organ at risk volume,: 3D: 3D distance.

organ movement via 2D films recorded every 3 minutes over a 15-minute period on an X-ray simulator. In a randomized phase 2/3 trial comparing 20 Gy in 5 fractions with 24 Gy in 2 fractions (SRT) for spinal metastases, no cases of radiation-induced myelopathy were documented using a 0.15–0.20 cm PRV margin for the spinal cord, with SRT showing superiority in pain relief [21]. This safety was achieved with the caution that an additional intrafractional scan for positioning was performed if the treatment length exceeded 30 minutes. This necessity for an additional intrafractional scan was confirmed in our study, which revealed that the intrafractional movement of the spinal cord (representing the patient's movement) significantly increased over time, with the mean intrafractional movement reaching 0.15 cm at the third MRI (Fig 1 and Table 3). In contrast, radiation-induced myelopathy occurred in 1 out of 86 patients when a 0 cm margin was used for the PRV and PTV via bony anatomical matching [22]. In a prospective trial comparing 8 Gy in 1 fraction with 16–18 Gy in 1 fraction (SRT) using a similar margin concept for the PRV and PTV, the incidence of spinal cord signal changes on MRI at 24 months after radiotherapy was reported to be 3.6% in the 16–18 Gy arm and 1.7% in the 8 Gy arm [23]. Although late spinal cord complications were not reported, superiority in pain relief with SRT was not obtained in the trial. Excessive shrinkage of the PRV margins could lead to results similar to those suggested by our study although the adapt-to-shape method was available in MRgART.

In MRgART, margins have been the focus of investigation primarily for PTV. The investigation of PTV margin via MRgART for pelvic oligometastases reported that only a 2 mm PTV margin achieved a minimum of 95% GTV coverage while reducing the dose to the bowel [24]. The validity of PTV margin reduction is often evaluated by GTV dose coverage via online planning MRI and posttreatment MRI (the first and third MR images, respectively, in this study) [25,26]. For example, margin assessments for rectal cancers have investigated the prescribed dose coverage of 95% of the GTV in 90% of patients [26]. In intracranial MRgART, a 3 mm PTV margin was recommended to cover 98% of the clinical target volume in 95% of the fractions in 95% of patients [27]. Although the standard to set the PTV margin for maintaining target dose coverage was not established, one of the standards would be 90% confidence of coverage of the clinical target volume by the 95% isodose, which used to create the Van Herk formula. As MR-Linac is a relatively new modality, more data are needed for the PTV margin and the PRV margin in MRgART.

The difficulty in reducing the PRV and PTV margins in MRgART lies in the challenge of minimizing treatment time. First, the adapt-to-shape method, which requires more time than the adapt-to-position method, is essential because the latter provides only plan shifts in the x-, y-, and z-axis directions (three degrees of freedom) [28]. The adapt-to-shape method ensures greater accuracy than the 6-axis correction in six degrees of freedom. Second, optimization recalculation is needed for each treatment fraction in MRgART. Third, Elekta Unity uses 7 MV flattening-filter-free photons, but the dose rate is fixed at 425 monitor units per minute [29]. Compared with 2400 monitor units per minute of conventional Linac, Elekta Unity requires longer treatment times [30]. Finally, Elekta Unity does not yet perform volumetric modulated arc therapy, which would shorten treatment times compared with intensity-modulated radiotherapy [31]. A comparison of beam-on times for MR-based IMRT using Co-60 sources, MR-Linac IMRT and volumetric modulated arc therapy via conventional Linac for spine SRT reported mean beam-on times ± SDs of 49.7± 11.1 minutes, 27.6±5.1 minutes and 4.0±1.1 minutes, respectively [32]. As a result, long treatment times are needed for MRgART. In this situation, proper optimization recalculation using the second MRI might contribute to reducing PTV margins if intrafractional movement is detected and corrected [26].

The PRV margins for the spinal cord and the spinal canal in this study were 0.11 cm and 0.11 cm, respectively. The results of the PRV margins are relatively smaller than the PRV margins from prospective trials of conventional Linac; however, these margins may appear insufficient for the y-axis (craniocaudal direction) [21]. This is because the formula is designed to avoid underestimating the high-dose components of serial structures in 90% of cases [12]. As a result, the effect of movement in the craniocaudal direction is considered relatively small due to the cylindrical shape of the spinal canal and spinal cord (Fig 2). Therefore, the calculated PRV margin is considered valid. However, it should be noted that the variations also reported that the multiplying value of sigma ranged from 1.2×Σ to 1.8×Σ and from −0.2×σ to 0.6×σ [33]. Further margin information, including 95% prediction intervals, was reported in this study. Relatively small prediction intervals were observed in the left, right, and anterior directions, all within ±0.11 cm. In the z-axis (vertical direction), the calculated margins in the posterior direction were larger than those in the anterior direction, suggesting that adding asymmetrical margins could be an option.

This study was subject to several limitations. Because MR-Linac is a relatively new modality, patient immobilization methods are still being developed. Proper body fixation, such as a vacuum cushion can reduce intrafractional movement [34]. Moreover, the retrospective nature of this study is associated with inherent limitations. The intervals between the first MRI and the second or third MRI varied, and the areas of the MRI scans also varied, leading to some missing data. In this study, SRT was performed in only 3 fractions. Using the margin formula in such cases is limiting because the formula assumes an infinite number of fractions [19].

In conclusion, the PRV margin for the spinal cord is necessary even if MRgART with the adapt-to-shape method is used. Based on the PRV margin formula and 95% prediction intervals, the 3D symmetric PRV margins are 0.11 cm and 0.26 cm, respectively, for the spinal cord and 0.11 cm and 0.25 cm, respectively, for the spinal canal. This study also suggested that directional preferences exist for movement; therefore, asymmetric PRV margins for the spinal cord and the spinal canal might be an option.

## Supporting information

**S1 File. Detailed patient information and three-dimensional coordinates of structures for each fraction.**
(XLSX)

## Acknowledgments

We thank to the radiation technologists at Tohoku University Hospital who contributed to the acquisition of MRI data.

## Author Contributions

**Conceptualization:** Takaya Yamamoto.

**Data curation:** Takaya Yamamoto, Shohei Tanaka.

**Formal analysis:** Takaya Yamamoto.

**Funding acquisition:** Takaya Yamamoto.

**Investigation:** Takaya Yamamoto, Shohei Tanaka, Noriyoshi Takahashi, Rei Umezawa, Yu Suzuki, Keita Kishida, So Omata, Kazuya Takeda, Hinako Harada, Kiyokazu Sato, Yoshiyuki Katsuta, Noriyuki Kadoya.

**Methodology:** Takaya Yamamoto.

**Project administration:** Takaya Yamamoto, Keiichi Jingu.

**Visualization:** Takaya Yamamoto.

**Writing – original draft:** Takaya Yamamoto.

**Writing – review & editing:** Takaya Yamamoto, Shohei Tanaka, Noriyoshi Takahashi, Rei Umezawa, Yu Suzuki, Keita Kishida, So Omata, Kazuya Takeda, Hinako Harada, Kiyokazu Sato, Yoshiyuki Katsuta, Noriyuki Kadoya, Keiichi Jingu.

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
