## [Decision Letter · Decision Letter 0]

6 Aug 2024

PONE-D-24-26533Investigation of intra-fractional spinal cord and spinal canal movement during stereotactic MR-guided online adaptive radiotherapy for kidney cancerPLOS ONE

Dear Dr. Yamamoto,

Thank you for submitting your manuscript to PLOS ONE. After careful consideration, we feel that it has merit but does not fully meet PLOS ONE’s publication criteria as it currently stands. Therefore, we invite you to submit a revised version of the manuscript that addresses the points raised during the review process.

We look forward to receiving your revised manuscript.

Kind regards,

Minsoo Chun, Ph.D.

Academic Editor

PLOS ONE

Journal Requirements:

KJ has received financial support from Elekta KK.

TY, ST, NT, RU, YS, KK, SO, KT, HH, KS, YK and NK have no conflicts of interests relevant to this work.

We note that one or more of the authors are employed by a commercial company: Elekta KK. 

“The funder provided support in the form of salaries for authors, but did not have any additional role in the study design, data collection and analysis, decision to publish, or preparation of the manuscript. The specific roles of these authors are articulated in the ‘author contributions’ section.”

**Additional Editor Comments:**

Both reviewers pointed out several English grammer errors.

Provide types of the MRgRT machine, and prescription information in Abstract.

There are several previous researches regarding need for PRV margin in MRgRT for other treament site, please search and add them in Introduction.

Provide CT scan information in M&M.

Arrange Table line for better understanding.

In Fig 1, label x and y axis, and mark with different shape to distinguish each point.

In Fig 2, describe what a, b, and c means.

Reviewers' comments:

Reviewer's Responses to Questions

**Comments to the Author**

1. Is the manuscript technically sound, and do the data support the conclusions?

Reviewer #1: Partly

Reviewer #2: Partly

2. Has the statistical analysis been performed appropriately and rigorously? 

Reviewer #1: Yes

Reviewer #2: Yes

3. Have the authors made all data underlying the findings in their manuscript fully available?

Reviewer #1: Yes

Reviewer #2: No

4. Is the manuscript presented in an intelligible fashion and written in standard English?

Reviewer #1: No

Reviewer #2: No

5. Review Comments to the Author

**Reviewer #1:** Authors investigated intrafractional spinal cord movement and it is well written and might be beneficial to medical physics community. Here are my comments.

Introduction section - " P3L52-53, " How shold RV margins be applied to organs?" please rephrase with proper grammar.

Method section - P4L74-77, " why is the letter size not uniform?

in P4L87, "The MR-Linac of this study was the Elekta Unity. please rephrase with proper grammar.

in P5L106, how did use DIR in Monaco, did you use VOI?

in P6L122, what do you mean by inter-fraction errors were modified?

in P6132-135, can you justify your method of considering Distance2 as interactional errors even though two MRIs are only 38 min apart?

Result section -

in P7L139, can you also compare with Van Herk's margin recipe (2.5 Big Sigma +0.7 little sigma)?

in Table2, please clarify what No. (57) is?

It seems redundant and excessive in data display between Table2-5, did you use DIR at all for manual contouring?

in P11L193-196 I cannot agree with author's speculation.

Discussion section - the 1.6mm motion of spine cord is from movement of patient or from cord itself inside of spinal canal?

**Reviewer #2: **The authors investigated the intrafractional movement of spinal cord and spinal canal during MR-guided stereotactic radiotherapy for kidney cancer. This study is of interest in the field of radiation oncology. However, it’s not ready for publication in its current state. The English need to be checked. A few details still need to be clarified, and the data need to be presented in a clearer manner.

Line 123. Explain what is “the average structure movement”. Is it an average of certain points in the structure, or all points in the structure?

Line 123-136. It’s hard to understand the calculations. Please check English to make them clear to understand. Is SD1 calculated for Distance 1 or Distance 2, or both? What does it mean “average movement between Distance 1 and Distance 2”. Similar questions to those in Secondly…, and Thirdly…

Line 139. PRV margin formula needs to be made easy to read, e.g., “Σ - intra2” change to (Σ-intra)2. Please provide more detail about the margin calculation model used and comment on its appropriateness for hypofractionated or SBRT situations.

Table 4-5, too many raw data tables of the distances. Suggest that authors summarize or just show/plot the difference of results using manual registration from those using deformable image registration. This can tell the accuracy of deformable image registration.

Table 2 and 3 and Figure 1 showed two patients had larger Y direction movement. Please explain or discuss the outliers?

Figure 2. please add space between images. also explain what are (A), (B), (C) presenting. Are they for different patients (specify patient No.). Please use the same contrast to show cord/canal and vertebra discs as in (B) so we can see craniocaudal shifts. Also suggest to show the spinal ord/canal contours on one of the images.

Table 6. suggest to calculate margins for each direction of X, Y, and Z. Authors mentioned that the shift was high in vertical direction, and the expectation of PRV margin reduction to 0cm. Authors also concluded at the end that “there are directional preference for movement”. So it’s better to provide margin in each direction, so we can know the minimal margin to apply in different direction when treating kidney cancer with MRgRT.

Line 266. Should it be “48.82±10.44 minutes and 3.95±1.13 minutes” for IMRT and VMAT, respectively?

Minor comments:

Suggest use 3D instead of “straight-line”.

Change “at the timing of” to “at the time of”.

Line 52: how “should” PRV margins…

Line 95: “post-treatment planning” should be “post-treatment”?

Line 134: was calculated “for” each “patient”…

Line 135: was calculated from all patients’ “” average data…

Line 255: “shift in the x, y, and z-axis direction” – why this is “four” degrees of freedom?

Line 269: suggest remove “unfortunately”

6. PLOS authors have the option to publish the peer review history of their article (what does this mean?). If published, this will include your full peer review and any attached files.

Reviewer #1: No

Reviewer #2: No

---

## [Author Response · Author response to Decision Letter 0]

20 Aug 2024

The following response was also provided in Response to Reviewers.docx 

file.

Response to additional editor comments:

Both reviewers pointed out several English grammer errors.

Response:

Thank you for your comments. We improved the English using professional proofreading. We enclosed certification of English professional proofreading.

Provide types of the MRgRT machine, and prescription information in Abstract.

Response:

Thank you for your advice. We added the information about MRgRT machine and prescription information in Abstract.

There are several previous researches regarding need for PRV margin in MRgRT for other treament site, please search and add them in Introduction.

Response:

Thank you for your suggestion. We searched for relevant papers and found several studies that discuss PRV margins in the context of PTV margin reduction in MRgRT for other treatment sites. After considering the overall structure and flow of the manuscript, we have added a discussion of these findings in the Discussion section.

Provide CT scan information in M&M.

Response:

Thank you for your advice. We added the information about CT scan in M&M.

Arrange Table line for better understanding.

Response:

Thank you for your suggestion. We reviewed the tables and made modifications to improve clarity and understanding.

In Fig 1, label x and y axis, and mark with different shape to distinguish each point.

In Fig 2, describe what a, b, and c means.

Response:

Thank you for your advice. We modified Fig 1 and 2.

Response to Reviewers

For Reviewer #1

Is the manuscript presented in an intelligible fashion and written in standard English?

Reviewer #1: No

Response:

Thank you for your comments. We improved the English using professional proofreading. We enclosed certification of English professional proofreading.

1) Introduction section - " P3L52-53, " How shold RV margins be applied to organs?" please rephrase with proper grammar. 

Response:

Thank you for your advice. We modified the sentence.

“How should PRV margins be applied to organs?”

2) Method section - P4L74-77, " why is the letter size not uniform?

Response:

Thank you for pointing this out. We have now standardized the letter size to ensure consistency throughout the manuscript.

3) in P4L87, "The MR-Linac of this study was the Elekta Unity.” please rephrase with proper grammar. 

Response:

Thank you for your suggestion. We modified the sentence.

“The MR-Linac used in this study was the Elekta Unity.”

4) in P5L106, how did use DIR in Monaco, did you use VOI?

Response:

Thank you for your question. We did not use VOI; instead, we utilized the “Adapt Anatomy” function in Monaco to align the first MRI with the second and third MRIs. In the “Adapt Anatomy” function, VOI cannot be used. While VOI could be applied if image fusion were used, but we did not use image fusion. If image fusion was used, we ensured that all values in the “Transformation Data” were reset to zero to accurately measure coordinate deviation. To clarify this, we have added the following explanation to the manuscript:

“All the “transformation data” values were subsequently reset to zero, and the delineations were adjusted to the second and third MR images via deformable image registration (DIR) through the “Adapt Anatomy” function in Monaco.”

5) in P6L122, what do you mean by inter-fraction errors were modified?

Response:

Thank you for question. The sentence meant that the inter-fractional structure changes (shift, rotation or deformation) were modified using ATS method. We modified the terms “inter-fraction errors” as follows:

“interfractional structure movement is modified at the time of the initial scan, effectively reducing interfractional setup margins to zero”

6) in P6L132-135, can you justify your method of considering Distance2 as interactional errors even though two MRIs are only 38 min apart?

Response:

Thank you for your question. As described in the Materials and Methods section, we based our approach on the methodology reported by Suzuki et al. (https://doi.org/10.1016/j.radonc.2006.03.006). In their study, they calculated intrafractional organ motions using an X-ray simulator with images taken only 15 minutes apart from the initial image. Therefore, we believe our method is valid, even with MRIs taken 38 minutes apart. To clarify this, we have added the following information to the Discussion section:

“Suzuki et al. reported PRV margins of 0.22–0.24 cm for the cervical vertebrae, which were calculated from interfractional and intrafractional errors [13]. They measured intrafractional organ movement via 2D films recorded every 3 minutes over a 15-minute period on an X-ray simulator”

7) Result section -

in P7L139, can you also compare with Van Herk's margin recipe (2.5 Big Sigma +0.7 little sigma)?

Response:

Thank you for your proposal. Yes, we can calculate the Van Herk's margin recipe. The formula added the results in method section and in the table 4.

8) in Table2, please clarify what No. (57) is?

Response:

Thank you for your comment. "No. (57)" of spinal canal refers to the 57 MRI series that included the scanning area from the upper edge of Th12 to the lower edge of L3 in both the first MRI and the second MRI. The spinal canal movement in these 57 MRI series was measured. We have added this clarification as a footnote in Table 2:

“†The number 57 indicates that 57 MRI series included the scanning area from the upper edge of Th12 to the lower edge of L3 in both the first and second MRIs, and the spinal canal movement in these 57 MRI series was measured.”

9) It seems redundant and excessive in data display between Table2-5, did you use DIR at all for manual contouring?

Response:

Thank you for your comments and advice. We used rigid registration for manual registration, as described in the method section. We have revised the text in the methods section to clarify this:

“To confirm the accuracy of the DIR, manually rigid registration with or without manual modification of structure contouring (manual registration) was also performed”

We agree that the tables were excessive, and have therefore moved Tables 4 and 5 to the supplemental materials.

10) in P11L193-196 I cannot agree with author's speculation.

Response:

Thank you for your advice. The speculation is not suitable for result section, therefore the sentence removed from figure legends.

11) Discussion section - the 1.6mm motion of spine cord is from movement of patient or from cord itself inside of spinal canal?

Response:

Thank you for your question. We believe the movement is primarily due to patient movement. This conclusion is supported by the high correlation coefficients between the 3D distances of the spinal cord and spinal canal at the second and third MRIs, which were 0.88 (p<0.01) and 0.92 (p<0.01), respectively. These high coefficients suggest that the movements of the spinal cord and spinal canal were closely related, likely reflecting overall patient movement rather than independent movement of the spinal cord within the canal. However, we cannot entirely rule out some intrinsic movement of the cord itself. We have added this information (the correlation coefficients) to the Results section.

For Reviewer #2

1) 3. Have the authors made all data underlying the findings in their manuscript fully available? Reviewer #2: No

A few details still need to be clarified, and the data need to be presented in a clearer manner.

Response:

Thank you for your comments. We apologize for the confusion caused by the previous Excel files. We have now uploaded clearer raw data, including the coordinate points. Thank you again for bringing this to our attention.

2) 4. Is the manuscript presented in an intelligible fashion and written in standard English? Reviewer #2: No

The English need to be checked.

Response:

Thank you for your comments. We improved the English using professional proofreading. We enclosed certification of English professional proofreading.

3) Line 123. Explain what is “the average structure movement”. Is it an average of certain points in the structure, or all points in the structure?

Response:

Thank you for your question. In Monaco, we determine a specific point within the structure using the 'center of structure' option. To clarify this, we have added the following sentence immediately before the sentence in question: 

“Three-dimensional coordinates of each structure were obtained from the “center of structure” option of Monaco; the coordinates consisted of the x-axis, y-axis and z-axis, which were the lateral axis, craniocaudal axis and vertical axis, respectively, with three-dimensional coordinates in centimeters.”

4) Line 123-136. It’s hard to understand the calculations. Please check English to make them clear to understand. Is SD1 calculated for Distance 1 or Distance 2, or both? What does it mean “average movement between Distance 1 and Distance 2”. Similar questions to those in Secondly…, and Thirdly…

Response:

Thank you for your question, and we apologize for the lack of clarity in our original explanation. We understand that this paragraph was complex, particularly regarding the calculations involving SD1, Distance1, and Distance2. With the assistance of English proofreading, we have revised this section to improve clarity: 

“1. 1. Intrafractional systematic error (Σ-intra): The average structural movements at each fraction were calculated by using the 3D distance from the first MRI to the second MRI (Distance1) and from the first MRI to the third MRI (Distance2). The mean ± standard deviation (SD) of these average movements across all fractions was calculated, with the resulting SD termed SD1, which represents Σ-intra (intrafractional systematic error).

2. Intrafractional random error (σ-intra): The mean ± SD between Distance1 and Distance2 at each fraction was calculated, and the SD for each fraction was determined (SD2). The root mean square of SD2 from all fractions was calculated, which was used as σ-intra, representing intrafractional random error.

3. Interfractional systematic error (Σ-inter): The average Distance2 for each patient was calculated from 3-fraction Distance2 data, and the mean ± SD of these averages across all patients was determined, with the resulting SD termed SD3. SD3 was used as Σ-inter, representing interfractional systematic error.

4. Interfractional random error (σ-inter): For each patient, the mean ± SD was calculated via 3-fraction Distance2 data (SD4) data. The root mean square of SD4 from all patients was calculated, which was used as the σ-inter, representing interfractional random error.”

5) Line 139. PRV margin formula needs to be made easy to read, e.g., “Σ - intra2” change to (Σ-intra)2. Please provide more detail about the margin calculation model used and comment on its appropriateness for hypofractionated or SBRT situations.

Response:

Thank you for your advice. We have revised the formula for clarity, making it easier to read. Additionally, we have provided more details about the margin calculation model in the Methods and Discussion sections. However, we acknowledge that applying this formula to hypofractionated or SBRT situations is a limitation. To address this, we also calculated the 95% prediction intervals and discussed this limitation in both the Discussion section and the limitations paragraph.

We have added the following discussion:.

“This inconsistency partly arises because the formula assumes an infinite number of fractions, which may not be applicable for MRgART with a small number of fractions [16,25]. To account for this assumption, we also examined the 95% prediction intervals, and the results were consistent with the PRV margin formula (Table 4).”

“In this study, SRT was performed in only 3 fractions. Using the margin formula in such cases is limiting because the formula assumes an infinite number of fractions [19].”

6) Table 4-5, too many raw data tables of the distances. Suggest that authors summarize or just show/plot the difference of results using manual registration from those using deformable image registration. This can tell the accuracy of deformable image registration.

Response:

Thank you for your advice. Another reviewer also pointed out the issue of having too many tables. Therefore, we have removed Tables 4 and 5 and included them as supplementary materials. To demonstrate the accuracy of deformable image registration (DIR), we calculated Pearson's correlation coefficients between the 3D distances obtained using DIR and those obtained through manual registration.

Method section: “To evaluate the correlation, Pearson's correlation coefficients were computed between the 3D distances calculated via DIR and those obtained via manual registration.”

Result section: “The correlation coefficients between DIR and manual registration via Distance1 and Distance2 were 0.76 (p<0.01) and 0.83 (p<0.01) for the spinal cord and 0.89 (p<0.01) and 0.88 (p<0.01) for the spinal canal, respectively.”

7) Table 2 and 3 and Figure 1 showed two patients had larger Y direction movement. Please explain or discuss the outliers?

Response:

Thank you for your question. As you pointed out, two patients exhibited larger movements. Upon review, we found that one patient experienced increasing back pain over time while in the supine position, and the other patient, although without complaints, had a body mass index (BMI) of 32, which may have influenced the movement. We have added this information to the Discussion section.

" As shown in Figure 1, two patients (No. 10 and No. 18) exhibited larger spinal cord movements. One patient reported increasing back pain over time while in the supine position, and the other patient, although asymptomatic, had a body mass index (BMI) of 32, which may have influenced the results."

8) Figure 2. please add space between images. also explain what are (A), (B), (C) presenting. Are they for different patients (specify patient No.). Please use the same contrast to show cord/canal and vertebra discs as in (B) so we can see craniocaudal shifts. Also suggest to show the spinal ord/canal contours on one of the images.

Response:

Thank you for your comments. We have addressed your suggestions by splitting Figure 2 into two separate figures: Figure 2 and Figure 3. Figure 2 now shows images from Patient No. 10, including contouring of the spinal cord and spinal canal. Unfortunately, it was challenging to maintain the same contrast for displaying the cord/canal and vertebrae discs due to the fat suppression used in T2-weighted images, as the contrast varies depending on the fat content of the discs. Figure 3 presents images from Patient No. 18 using a checkerboard view for comparison.

9) Table 6. suggest to calculate margins for each direction of X, Y, and Z. Authors mentioned that the shift was high in vertical direction, and the expectation of PRV margin reduction to 0cm. Authors also concluded at the end that “there are directional preference for movement”. So it’s better to provide margin in each direction, so we can know the minimal margin to apply in different direction when treating kidney cancer with MRgRT. 

Response:

Thank you for your advice. We calculated the margins for X, Y and Z directions, and the result was added in the table.

10) Line 266. Should it be “48.82±10.44 minutes and 3.95±1.13 minutes” for IMRT and VMAT, respectively?

Response:

Thank you for your pointing this out. Thie sentence meant exactly what you mentioned. We modified this sentence.

“A comparison of intensity-modulated radiotherapy of the Co-60 MR-Linac system with volumetric modulated arc therapy of conventional Linac for spine SRT reported mean beam-on times ± SD of 48.82±10.44 minutes and 3.95±1.13 minutes, respectively.”

11) Suggest use 3D instead of “straight-line”.

Response:

Thank you for your advice. We modified this point using “3D distance” instead of “straight-line”.

12) Change “at the timing of” to “at the time of”.

Line 52: how “should” PRV margins…

Line 95: “post-treatment planning” should be “post-treatment”?

Line 134: was calculated “for” each “patient”…

Line 135: was calculated from all patients’ “” average data…

Line 269: suggest remove “unfortunately”

Re

---

## [Decision Letter · Decision Letter 1]

20 Sep 2024

PONE-D-24-26533R1Investigation of intrafractional spinal cord and spinal canal movement during stereotactic MR-guided online adaptive radiotherapy for kidney cancerPLOS ONE

Dear Dr. Yamamoto,

Thank you for submitting your manuscript to PLOS ONE. After careful consideration, we feel that it has merit but does not fully meet PLOS ONE’s publication criteria as it currently stands. Therefore, we invite you to submit a revised version of the manuscript that addresses the points raised during the review process.

We look forward to receiving your revised manuscript.

Kind regards,

Minsoo Chun, Ph.D.

Academic Editor

PLOS ONE

Additional Editor Comments :

I agree with both reviewers, but the grammar needs clarification.

In Figure 1, for example, there are redundant expressions regarding "first," "second," and "third," which can be interpreted as fractions or scans. Additionally, I find it difficult to agree that the range of plots in Figure 1B is narrower than those in 1A. Please clarify these points.

I also recommend that the figure caption be placed below the figure.

Reviewers' comments:

Reviewer's Responses to Questions

**Comments to the Author**

1. If the authors have adequately addressed your comments raised in a previous round of review and you feel that this manuscript is now acceptable for publication, you may indicate that here to bypass the “Comments to the Author” section, enter your conflict of interest statement in the “Confidential to Editor” section, and submit your "Accept" recommendation.

Reviewer #1: All comments have been addressed

Reviewer #2: (No Response)

2. Is the manuscript technically sound, and do the data support the conclusions?

Reviewer #1: Yes

Reviewer #2: Partly

3. Has the statistical analysis been performed appropriately and rigorously? 

Reviewer #1: Yes

Reviewer #2: No

4. Have the authors made all data underlying the findings in their manuscript fully available?

Reviewer #1: Yes

Reviewer #2: Yes

5. Is the manuscript presented in an intelligible fashion and written in standard English?

Reviewer #1: No

Reviewer #2: No

6. Review Comments to the Author

Reviewer #1: Thank you for the authors' hard work on the revision and responses. The revised manuscript looks much improved and clean. However, English grammar seems still need to be corrected in the entire manuscript. for example,

P3L46 - " planning the organ as risk volume"

P3L61 - " INevertheless"

P4L71 - "PRV margin.."

P19L346 - A Moreover

P19L352 - In conclusion, ta PRV...

And these are minor comments as below.

P5L97 - please describe the detail of T2 MRI protocol used.

As you said in the discussion P17L307, Van Herk's model is calculated for 90% confidence of coverage of CTV by 95% isodose. so, basically 90% of pt population will receive 95% of CTV dose with that margin formular. I have 3 IMPORTANT Qs since this is your key results.

1. How about the formular you used 1.3xbigSigma + 0.5xlittleSigma in P8L160? if we use that margin, what statistical meaning does this formular have to do with CTV coverage and pt population?

2. why are there a big discrepancy between 95% prediction intervals (0.26cm) and PRV margin (0.16) from the formular in Table 4?

3. it is hard to understand the rational ("inconsistency") regarding why you chose 95% prediction instead of Van Herk's formular on P17L308-312.

P9L196-197 I still do not agree with authors' interpretation of the results and significance of it.

P10L201-105 please describe more detail like larger movement of two patients and meaning of BMI 32. you mean larger pt tends to move more?

Throughout the paper, please clarify if the motion comes from patient's motion or spinal cord motion itself inside of spinal canal. You mentioned that you saw some specific cord motion, please describe in detail.

Unusually the Figure captions are located in the top of the figures, and it seems confusing, please check the PLOS ONE standard.

P18L325 - 48.82min is beam-on time or total treatment time?

P18L336-338 - it is hard to follow the meaning and significance of these sentences.

In discussion section, it was confusing because "PRV margin" and "PTV margin" were mentioned interchangeably. please clarity them in entire section for readers' better understanding. Thanks.

Reviewer #2: English is better than last version. A few more details still need to be clarified.

Major comments:

Line 143-158: both Distance1 and Distance2 are intrafractional errors. Why there are interfractional systematic error (3) and interfraction random error (4) calculated here (no interfractional structure movement considered in this study as mentioned in line 132-133)? To my understanding, these errors in 3 and 4 are standard intrafractional systematic and random errors calculated using Distance2, which is larger than Distance1. The errors calculated in 1 and 2 are not clear to me. I guess the authors wanted to use the average of Distance1 and Distance2 to calculate the systematic and random error, but I don’t see these errors are for each patient or for all patients. In my opinion, there are only intrafractional errors should be calculated and authors can use either Distance2 only (most conservative), or Distance1 only, or average of Distance1 and Distance2. Combining 1 and 2 with 3 and 4 (formula in line 160 and 163) is not reasonable to me, because they are both intrafractional errors and Distance2 is used in both pairs, thus may overestimate the margins. The authors should provide the reference for error calculation method in 1 and 2, or explain the method if they developed it.

Line 182-193: I think the movement data with manual registration are not needed to be included in this manuscript, since the correlation of DIR and manual are presented in lines 189-193.

Line 196-197: “the absolute values tended to increase from Figure 1A to Figure 1B, whereas the scatter of the plots per patient tended to decrease from Figure 1A to Figure 1B”. I only see 8/22 patients had decreased scatter.

Line 199: define “dispersion” of the 3D distance.

Line 201-203: specify the patient # who reported back pain, and patient who had body mass index of 32.

Line 246: Table 4 listed the PTV margin, but never mentioned PTV motion data.

Line 292 paragraph: PTV margin is not a topic in this study.

Line 354-355: the PRV margins for spinal cord are numbers (0.16cm and 0.26cm), but for spinal canal are ranges (0.16-0.19cm and 0.25-0.27cm)?

Minor comments:

Figure legends should be below the Figures.

Line 29: change “postplan” to “posttreatment” since you use “posttreatment” in the other places.

Line 61: Correct spelling “Nevertheless”.

Line 198: what dose this mean: “The mean 3-fraction mean 3D distance”?

Line 346: remove “A” before “Moreover”.

Line 352: correct “ta PRV”. Do you mean “The PRV”?

7. PLOS authors have the option to publish the peer review history of their article (what does this mean?). If published, this will include your full peer review and any attached files.

Reviewer #1: No

Reviewer #2: No

---

## [Author Response · Author response to Decision Letter 1]

24 Sep 2024

We have enclosed the responses to Reviewer 1, Reviewer 2, and the Editor in the 'Revised Manuscript with Track Changes.docx' file.

---

## [Decision Letter · Decision Letter 2]

30 Sep 2024

Investigation of intrafractional spinal cord and spinal canal movement during stereotactic MR-guided online adaptive radiotherapy for kidney cancer

PONE-D-24-26533R2

Dear Dr. Yamamoto,

We’re pleased to inform you that your manuscript has been judged scientifically suitable for publication and will be formally accepted for publication once it meets all outstanding technical requirements.

Kind regards,

Minsoo Chun, Ph.D.

Academic Editor

PLOS ONE

Additional Editor Comments (optional):

I accept this manuscript for publication.

Minor comments should be corrected in the final submission.

1. Unit in Table 2-4.

2. Figure 1: Plot should be larger (or fill and blank) for better visualization, and I think legend is necessary.

Reviewers' comments:

Reviewer's Responses to Questions

**Comments to the Author**

1. If the authors have adequately addressed your comments raised in a previous round of review and you feel that this manuscript is now acceptable for publication, you may indicate that here to bypass the “Comments to the Author” section, enter your conflict of interest statement in the “Confidential to Editor” section, and submit your "Accept" recommendation.

Reviewer #1: All comments have been addressed

Reviewer #2: All comments have been addressed

2. Is the manuscript technically sound, and do the data support the conclusions?

Reviewer #1: Yes

Reviewer #2: Yes

3. Has the statistical analysis been performed appropriately and rigorously? 

Reviewer #1: Yes

Reviewer #2: Yes

4. Have the authors made all data underlying the findings in their manuscript fully available?

Reviewer #1: Yes

Reviewer #2: Yes

5. Is the manuscript presented in an intelligible fashion and written in standard English?

Reviewer #1: Yes

Reviewer #2: Yes

6. Review Comments to the Author

Reviewer #1: (No Response)

Reviewer #2: Overall, all comments were addressed, and English was corrected. Please add unit "cm" for data in Table 2-4. No other comments.

7. PLOS authors have the option to publish the peer review history of their article (what does this mean?). If published, this will include your full peer review and any attached files.

Reviewer #1: No

Reviewer #2: No

---

## [Editor Report · Acceptance letter]

21 Oct 2024

PONE-D-24-26533R2 

PLOS ONE

Dear Dr. Yamamoto, 

I'm pleased to inform you that your manuscript has been deemed suitable for publication in PLOS ONE. Congratulations! Your manuscript is now being handed over to our production team.

Kind regards, 

on behalf of

Dr. Minsoo Chun 

Academic Editor

PLOS ONE